# Tumor Infiltrating Lymphocytes in Multi-National Cohorts of Ductal Carcinoma In Situ (DCIS) of Breast

**DOI:** 10.3390/cancers14163916

**Published:** 2022-08-13

**Authors:** Sunil S. Badve, Sanghee Cho, Xiaoyu Lu, Sha Cao, Soumya Ghose, Aye Aye Thike, Puay Hoon Tan, Idris Tolgay Ocal, Daniele Generali, Fabrizio Zanconati, Adrian L. Harris, Fiona Ginty, Yesim Gökmen-Polar

**Affiliations:** 1Department of Pathology and Laboratory Medicine, Emory University School of Medicine, Atlanta, GA 30322, USA; 2Winship Cancer Institute, Atlanta, GA 30322, USA; 3GE Global Research Center, Niskayuna, NY 12309, USA; 4Center for Computational Biology and Bioinformatics, Department of Biostatistics and Health Data Science, School of Medicine, Indiana University, Indianapolis, IN 46202, USA; 5Anatomical Pathology, Singapore General Hospital, Singapore 169856, Singapore; 6Division of Pathology, Singapore General Hospital, Singapore 169856, Singapore; 7Department of Laboratory Medicine and Pathology, Mayo Clinic Arizona, Phoenix, AZ 85054, USA; 8Department of Medical, Surgery and Health Sciences, University of Trieste, 34127 Trieste, Italy; 9Cancer and Haematology Centre, Department of Oncology, Oxford University, Oxford OX3 7LE, UK

**Keywords:** breast cancer, ductal carcinoma in situ, ethnicity, tumor infiltrating lymphocytes

## Abstract

**Simple Summary:**

Tumor-infiltrating lymphocytes (TILs) are prognostic in invasive breast cancer. However, their prognostic significance in ductal carcinoma in situ (DCIS) has been controversial. We used different scoring methods for TILs in multi-national cohorts from Asian and European women. Stromal TILs, touching TILs, circumferential TILs, and hotspots were quantified on H&E-stained slides and correlated with the development of second breast cancer events (BCE) and other clinico-pathological variables. Older women, hormone receptor positivity, and the presence of circumferential TILs were weakly associated with the absence of BCE at 5-year follow-up in all cohorts. In multivariable analysis, older women with circumferential TILs were less likely to develop BCEs (Wald test *p* = 0.01). Asian patients were younger with larger, higher grade, HR negative DCIS lesions, and higher TIL variables. The spatial arrangement of TILs may serve as a better prognostic indicator in DCIS cases than stromal TILs alone.

**Abstract:**

Tumor-infiltrating lymphocytes (TILs) are prognostic in invasive breast cancer. However, their prognostic significance in ductal carcinoma in situ (DCIS) has been controversial. To investigate the prognostic role of TILs in DCIS outcome, we used different scoring methods for TILs in multi-national cohorts from Asian and European women. Self-described race was genetically confirmed using QC Infinium array combined with radmixture software. Stromal TILs, touching TILs, circumferential TILs, and hotspots were quantified on H&E-stained slides and correlated with the development of second breast cancer events (BCE) and other clinico-pathological variables. In univariate survival analysis, age older than 50 years, hormone receptor positivity and the presence of circumferential TILs were weakly associated with the absence of BCE at the 5-year follow-up in all cohorts (*p* < 0.03; *p* < 0.02; and *p* < 0.02, respectively, adjusted *p* = 0.11). In the multivariable analysis, circumferential TILs were an independent predictor of a better outcome (Wald test *p* = 0.01), whereas younger age was associated with BCE. Asian patients were younger with larger, higher grade, HR negative DCIS lesions, and higher TIL variables. The spatial arrangement of TILs may serve as a better prognostic indicator in DCIS cases than stromal TILs alone and may be added in guidelines for TILs evaluation in DCIS.

## 1. Introduction

Recent studies have clearly documented the role of immune cells (tumor infiltrating lymphocytes (TILs)) in human cancers, including breast cancer [1,2,3,4]. These studies show that increased numbers of lymphocytes are associated with an improved recurrence free survival (RFS) and overall survival (OS) in patients treated with standard of care chemotherapy with HER2+ and triple negative breast cancer. These associations have been predominantly observed with stromal TILs and not with intraepithelial TILs. Immune infiltration in tumors could be due to the presence of novel antigens generated by tumor associated mutations and/or could be due to the destruction of the stromal structures and local tissue damage. The study of invasive tumors is complicated due to factors such as (immune) response to invasion-related tissue damage. This can be potentially further modified by effects of prior neo-adjuvant therapy. Alternatively, the immune response in ductal carcinoma in situ (DCIS) is likely to be almost entirely due to a reaction to tumor neoantigens.

The immune response to an antigen is determined by a number of factors including the human leukocyte antigens (HLA). The Polyak group [5] has recently analyzed six cases of matched DCIS and invasive carcinomas and found MHC-1 associated differences in immune signaling. As HLA show linkage disequilibrium, ethnic differences could contribute to observed differences in the immune responses to early stage cancers. These have been reported in a very limited number of studies [6,7]. Ethnic/racial differences also contribute to a number of processes including the host’s response to infection, cytokine and chemokine mediated inflammation, and toll receptor signaling pathway [8,9]. Furthermore, ethnicity-based differences in outcomes have been documented in DCIS [10].

Ductal carcinoma in situ (DCIS) accounts for 20–25% of the cases of breast cancer [11]. DCIS is primarily treated by surgical excision. Patients with DCIS are at an increased risk of developing recurrent DCIS or invasive carcinoma. The incidence of ipsilateral disease recurrence is approximately 30% in patients treated with a lumpectomy. Clinical trials have documented that this risk for recurrent disease can be reduced by approximately 50% with the addition of radiation [12]. These data have resulted in mastectomies or surgical excisions plus radiation as the standard of care for the management of DCIS [12]. Tamoxifen is also offered for patients with ER+ DCIS [13].

Immune response in DCIS, in the form of tumor infiltrating lymphocytes (TILs), has been evaluated in a number of small and large single institutional studies [14,15,16,17,18,19,20,21,22]. These have shown a significant association between the presence of TILs and poor prognostic features of DCIS such as high grade, necrosis, absent ER expression, and positive HER2 status and dense TILs. In some studies, the presence of TILs is associated with a risk of recurrence [14,15,16]. More specifically, the assessment of stromal TILs, as detailed by the International Immuno-Oncology Biomarker Working Group, did not demonstrate an association with DCIS recurrence. Toss et al. [23], in a large single institutional study, analyzed the immune response to DCIS using a number of different methods. In addition to the classical stromal TILs, they also analyzed the distribution of the TILs and their relationship with the epithelial component. They categorized TILs as being circumferentially distributed around the ducts, whether they were “touching” the epithelial cells and whether there were hotspots; each of these findings were further subjectively assessed. Their analysis showed that touching TILs in DCIS was an independent prognostic variable in DCIS [23]. We adopted these methods to analyze a series of DCIS cases from four institutions and found that the presence of TILs was associated with incidences of recurrence [24]. Recent data indicates immunological differences in invasive cancer in Asians, but less is known about DCIS [25,26] The current study focuses on the assessment of TILs in a multi-national and multi-ethnic cohort to seek to understand the impact, if any, of TILs on the likelihood of development of invasive ipsilateral breast cancer events.

## 2. Materials and Methods

### 2.1. Clinical and Histological Data

After the appropriate institutional approvals, histologically confirmed cases of DCIS were obtained from the pathology databases of Oxford University (Oxford, UK), Singapore General Hospital, (SGH) Singapore, Trieste University Hospital (Trieste, Italy), and Mayo Clinic, Arizona. All cases had to have either a history of development of a second breast cancer event or a minimum of 5 years without any additional breast event. All cases were de-identified before analyses. A multi-institutional cohort of 266 cases, of which 70 had developed second breast cancer events, was generated; 10 patients were excluded as adequate information was not available. One hematoxylin and eosin (H&E) stained slide from each of the cases of DCIS from Mayo and Italian cohorts was reviewed. The analysis of Singapore cases was made on 2mm tissue microarray (TMA; 3 per case) sections. The analysis of the Oxford cases was performed on TMA sections.

The sections were assessed for the presence of tumor-infiltrating lymphocytes (TILs) by two observers (SB and YGP) and the discrepancies resolved by discussion. The observers were blinded to the clinical data. The amount and distribution of TILs were assessed using the previously described methodologies, including the estimation of stromal TILs, the estimation of TILs touching DCIS ducts, and the assessment of circumferential TILs and presence or absence of hotspots [23,24]. TILs associated with areas of necrosis, prior biopsy, and crush artefacts were excluded from the analysis. The clinical parameters of individual cohorts are detailed in Appendix A. Briefly, the stroma in the vicinity of DCIS was assessed for the percentage occupancy by TILs. Similar to previous publications [14], the percentage of stromal TILs was characterized as low (<=5%) or high (>5%). The distribution of TILs around the duct was further assessed as circumferential or not, based on whether there was circumferential or near-circumferential (>75% of circumference) cuffing of DCIS duct by lymphocytes or plasma cells [17]. The presence of a cluster of lymphoid cells (>50 cells) indicated the presence of a hotspot. TILs were categorized as touching if they were either touching or within one lymphocyte cell thickness from the DCIS duct basement membrane. The mean number of TILs per DCIS duct was calculated and divided into three categories: absent very scanty (≤5 touching TILs per duct), sparse (6–20 touching TILs per duct), and dense (>20 touching TILs per duct). For statistical analysis, absent, and sparse TILs were considered as absent, and dense touching TILs were classified as present. As per the International Immuno-Oncology Biomarker Working Group guidelines, cases of papillary DCIS were excluded from analysis.

### 2.2. Ancestry Analysis

Self-reported ethnicity, in addition to tissue samples for ancestry analysis, were available from Italian, Singapore, and Mayo cohorts. DNA was isolated from FFPE Samples using QIAGEN Allprep DNA/RNA FFPE Kit. After the DNA extraction, we used the Illumina FFPE QC Kit (real-time PCR assay) to evaluate the quality of DNA samples. Extracted FFPE samples that passed the QC test were processed for restoration using the Infinium HD FFPE Restore Kit. The protocol restored degraded FFPE DNA to an amplified state. The DNA was analyzed using the QC Infinium array assay using Roswell Park Genomic Shared Resource. The data was analyzed using an R implementation of the ADMIXTURE software [27] for individual ancestry inference, called ‘radmixture’. Specifically, ADMIXTURE is a software tool for maximum likelihood estimation of individual ancestries from multi-locus SNP genotype datasets. The 753 SNP markers that overlapped with 166,255 SNPs used in the K13 reference model were used for ancestry mapping.

### 2.3. Statistical Analysis

Associations of the patient age, DCIS size, nuclear grade, necrosis, estrogen and progesterone receptor status of touching TILs, stromal TILs, hotspots, and circumferential TILs were examined using Fisher’s exact tests with FDR = 0.1. The same approach was applied to evaluate if there was any difference in TIL scores in the individual cohort. Cox proportional hazard model [28] with FDR = 0.1 was used to evaluate the univariable associations of clinical outcome (second breast cancer event within 5 years) with clinical, histological, and TILs scoring. All p-values were adjusted using Benjamin-Hochberg procedure for multiple testing.

The stepwise variable selection procedure, the method of choice for identifying predictive variables in an unbiased manner, was used [29,30,31,32,33,34,35]. This heuristic procedure consists of a series of alternating forward selection and backward elimination steps, and ultimately selects a subset of variables to be included in the Cox-PH model. Stepwise variable selection procedure using My.stepwise.coxph function in R (with SLE = 0.25, and SLS = 0.25) was applied to obtain the best candidate model to associate with 5 years recurrence.

Since all patients from the Singapore cohort were Asian (albeit of Chinese, Malay, and Indian), while a high portion of Mayo/Italian cohort was non-Asian, we further sought to analyze cohort differences by studying the impact of ancestry. We compared clinical, histologic parameters, and TILs scoring in Asians among different cohorts to examine if there were any cohort differences.

## 3. Results

### 3.1. Analysis for Tumor-Infiltrating Lymphocytes (TILs) in Multi-Ethnic DCIS Cases

The study population whole sections were from 2 cohorts (Mayo, Arizona, and Trieste, Italy; *n* = 114) and TMA sections from 2 additional cohorts (Singapore and Oxford, UK; *n* = 142). Self-reported race was available in the Singapore cohort (63 Chinese, 1 Indian, 1 Malay, and 9 others); Mayo cohort (63 white, 1 Indian, and 1 Chinese), and Trieste cohort (all white). Approximately 24% of patients were below 50 years of age and 63% had DCIS size less than 20 mm. The grade of DCIS was high in 50% patients, intermediate in 37% patients, and low in 13% patients. Expression of hormone receptors (ER or PR) was noted in 81% of cases. A total of 27% of patients had been treated with a mastectomy and the rest were treated with a combination of lumpectomy, radiation, and hormone therapy. All patients had negative margins of resection, but the exact definition of margins varied amongst the cohorts and even within cohorts over the duration of the study. Follow-up data was available for all patients; this ranged from 6 months to 18 years. In total, 26.2% of patients had developed second breast cancer events.

#### 3.1.1. Combined Analysis of All Cohorts

The analysis of the stromal TILs showed that 60.5% of all cases were scored as having greater than 5% stromal TILs, whereas 37.5% of all cases formed small hotspots (aggregates of >50 cells) (Table 1). In addition, 75 (29.3% of total) cases showed periductal lymphocytic infiltration. Touching TILs were categorized as absent/sparse (≤5 touching TILs per DCIS duct) in 83.6% cases, and dense (>20 touching TILs per DCIS duct) in 16.4 % of cases.

#### 3.1.2. Analysis of TILs in Individual Cohorts

There was no difference in the distribution of stromal TILs in combined data when compared to individual cohorts (adjusted *p* = 0.421). Touching TILs were categorized as absent/sparse in 93.6 % of Italian, 88.1 % of Mayo, 78.3% of Oxford, and 78.1% of Singapore (adjusted *p* = 0.077) (Appendix A). Among the four cohorts, three cohorts represented very similar frequencies as the presence of circumferential TILs ranging between 20.9–29.8%, whereas the Singapore cohort had a higher percentage than the other cohorts (42.5%) with adjusted *p* = 0.044. The Singapore cohort had a higher frequency of stromal hotspots (54.8%) compared to the other cohorts (21.3% of Italian, 28.4% of Mayo, and 39.1% of Oxford) with adjusted *p* = 0.004.

#### 3.1.3. Correlation of TILs Assessments with Clinical and Histologic Parameters

Figure 1 shows the correlation plot of the assessments for all TILs scoring methods in relation to race/ethnicity and clinico-pathological parameters. Accordingly, Asian ethnicity (mostly Singapore) cohort compared to the three European ethnicity (Italian and Mayo) cohorts showed that Asian ethnicity tended to have higher scores with all TIL scoring methods in the following order for the presence of hotspots (adjusted *p* < 0.001), circumferential TILs (adjusted *p* = 0.064), touching TILs (adjusted *p* = 0.088), and stromal TILs was not different in Asians (*p* = 0.227). In addition, Asian patients were younger in age (adjusted *p* < 0.001) and had a larger tumor size (adjusted *p* = 0.022). Hormone receptor (HR) expression was not different in Asians. However, the presence of HR was positively correlated with stromal (adjusted *p* = 0.01), circumferential TILs (*p* = 0.015), and hotspots (*p* = 0.02) in overall population.

#### 3.1.4. Correlation of TILs Assessments with Clinical Outcome

Cox proportional hazard models were developed to determine the association of nine variables (ancestry, age, size, grade, hormone receptor (HR), stromal lymphocyte, touching TILs, circumferential TILs, and hotspot) with outcomes at 5-year follow-up. As both shown in Figure 2A,B, three variables were weakly associated with 5-year second breast cancer events including age (*p* = 0.02; adjusted *p* = 0.11), circumferential TILs (*p* = 0.03; adjusted *p* = 0.11), and hormone receptor status (*p* = 0.02; adjusted *p* = 0.11). Survival analysis in individual cohorts by each of the TIL’s parameters is shown Appendix A.

In multivariable analysis, stepwise selection on Cox proportional hazard model were applied to 7 variables (age, size, grade, lymphocyte, touching TILS, circumferential TILs, hotspot) (Figure 3). The combination of three variables—age, grade, and circumferential TILs (Wald test *p* = 0.02) predicted the likelihood of a second breast cancer event. 

#### 3.1.5. Impact of Ethnicity

DNA for ethnicity determination was available for all but the Oxford cohorts. The K13 model was chosen for our analysis, as this provided more relevant information regarding the distribution of population in our cohorts. These analyses confirmed that the Singapore cases were mainly Southeast Asian origin (47%) with an admixture of Mediterranean (10%) and North European (21%). On the other hand, the Mayo cohort resulted in 47% North-European mixed with 29% Mediterranean and 9% West-Asian. In contrast, the Italian cohort comprised of North-European (35%), Mediterranean (28%), and 12% West-Asian.

As the largest ethnic group was Asian, we sought to further analyze the association of ancestry with TILs and recurrence. Asian ethnicity was associated with a younger age at diagnosis, a larger size, and a greater incidence of tumor infiltrating lymphocytes irrespective of the assessing methodology. Further exploratory analysis of Asians from Singapore cohort in comparison with Asians from other cohorts revealed no significant differences in incidence of variables or association with TILs or second breast cancer event(s) (Appendix A). In spite of this, Asians had numerically higher event rate (Appendix A).

## 4. Discussion

The association of tumor-infiltrating lymphocytes (TILs) with improved prognosis in invasive breast cancer, particularly HER2 and triple negative subtypes, led to an assumption that TILs assessment of H&E samples can provide a similar prognostic value in DCIS. However, the relationship between TILs and DCIS outcomes is not straight forward [20,21,22,36,37,38]. Many studies have used the International Working TILs Group guidelines [1] to evaluate the percentage of stromal TILs and reported little to no association with DCIS recurrence. One of the reasons for these discrepant results is due to the variability of the endpoints. These endpoints include any recurrence, ipsilateral recurrence, and development of invasive carcinoma. Furthermore, most studies are not multi-centric and instead are based on relatively small cohorts. More importantly, the event rates are relatively low in most studies, which hampers the stability of the statistical methods.

The standard of care for treatment of DCIS consists of mastectomy or lumpectomy plus radiation. The patient population in the current study had been treated with variable modalities; this is well reflected in our cohorts. All patients from Singapore had been treated with mastectomy, while other cohorts had been managed by a mix of mastectomy and lumpectomy plus radiation. In the combined analysis of the cohorts, differences in treatment did not emerge as an important confounding factor. The need for follow-up information undoubtedly led to a bias in the number of events observed. The Oxford cohort had a significantly higher event rate (39.1%) than the other cohorts. However, after the merging of the cohorts, we observed a recurrence of 26.4%. This higher incidence of events could lead to the identification of associations that were not observed in some of the prior studies.

Knopfelmacher et al. [39] had previously demonstrated significant association with circumferential TILs and high ODX DCIS scores in a study of 46 cases. Toss et al. [23] assessed the impact of seven TILs scoring methods using a training and validation cohort of DCIS with long-term follow-up data in a large single institutional study. Among these scoring methods, dense touching TILs have been shown as an independent prognostic variable in DCIS rather than stromal TILs, which have been used in most of the earlier studies. In a study of 198 cases with recurrence follow-up reported, a significant association between circumferential TILs and worse disease-free survival [40]. Recently, Komforti et al. [24] assessed the pattern of TILs in a series of 97 cases by three different methodologies [20,23,41]. Stromal TILs showed the lowest concordance rate, followed by touching TILs by all pathologists. On the other hand, the assessment of circumferential TILs had a greater concordance rate. This study showed significant associations of touching TILs and circumferential TILs with the intermediate/high ODX DCIS score, but not stromal TILs. In contrast to these studies, we found that circumferential TILs were associated with a good prognosis, but no significant association was found for other TILs variables, including touching TILs and stromal TILs. The differences in these compared to our study may be attributed to the distribution of other clinico-pathological parameters including age, tumor size, grade, or hormone receptor status [40].

A major impact of cohort size and location was observed in the analysis of the association of TILs with recurrence. Circumferential TILs were not significant in any of the independent cohorts. However, their presence was associated with a decreased risk of recurrence in the combined analysis (*p* = 0.03). The other TILs parameters were not significant in individual cohorts or in the combined analysis. Race/ethnicity can be another factor that may affect the DCIS outcome. Analyzing 18 NCI-SEER Registries, Lui et al. reported that black and Hispanic women had higher ipsilateral breast tumor (IBT) risk when compared to white women [10,42]. The risk for contralateral breast tumors (CBT) was also significantly high in black and Asian/Pacific Islander (PI) women compared with white women (*p* heterogeneity < 0.0001). While the black women older than 50 years and with comedo necrosis were more likely to have CBT, Asian/PI women younger than 50 years and with lack of comedo DCIS had more significantly associated with CBT [10]. These results were consistent with our study that Asian women were younger in age and had significantly higher recurrence rate than the European white cohort. The correlation with ethnicity with hormone receptor status was not analyzed due to large portion of missing data.

The current study is one of the larger studies to assess the impact of TILs in DCIS. In spite of this, there are several limitations. Part of the study was performed on TMAs as this was the only material available for analysis. However, Toss et al. [23] have previously documented a high degree of correlation between TILs observed in TMAs and in whole sections. Another limitation associated with retrospective multi-institutional studies is the variability in assessment of margin status and treatment regimens. Furthermore, the requirement of follow-up data could have led to a bias in selection of the patients for the study. This may have resulted in higher event rates observed in the study. The latter was a strength as well as a limitation of the study. The higher events rate could have made it possible to define associations not observed in prior studies, which had low event rates. Lastly, differences have been documented in risk of subsequent invasive carcinoma in Asian women of different geographic origin [43]. The current study had too few women from different Asian origins to tease out more information regarding this parameter.

## 5. Conclusions

In conclusion, our study and others provide promising results to include the different TILs scoring methods to the current guidelines for predicting the prognostic significance of TILs in DCIS.

## Figures and Tables

**Figure 1 cancers-14-03916-f001:**
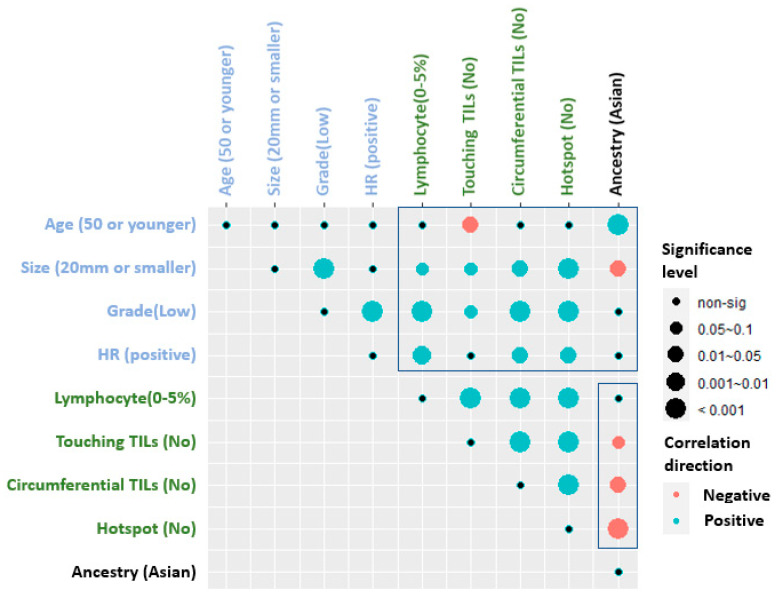
Correlation analysis of tumor-infiltrating lymphocytes (TILs) assessments with clinical and histologic parameters. The dependence of each pair was evaluated by Fishers exact test, and adjusted *p* (FDR) = 0.1 was applied to 36 pairs; *p*-value was adjusted using Bonferroni-Holm procedure; Blue and red circles represent positive and negative correlations, respectively; Significance levels were shown based on the size of the circles.

**Figure 2 cancers-14-03916-f002:**
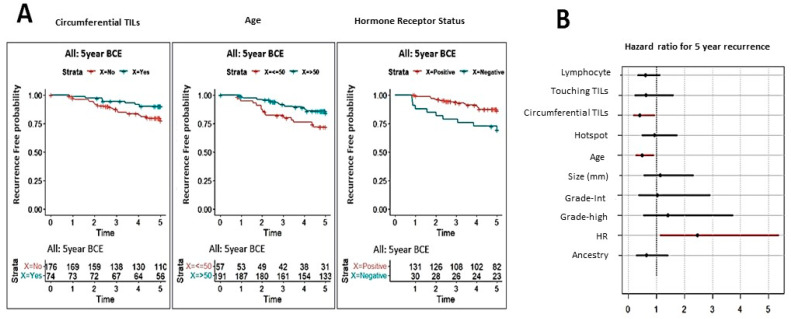
Correlation analysis of tumor-infiltrating lymphocytes (TILs) assessment with clinical outcome; (**A**) Kaplan-Meier (KM) plots for 5-year recurrence-free survival according TILs assessments and clinical parameters (univariable analysis). The significant associations were shown in the figure; (**B**) A forest plot showing the hazard ratio and 95% confidence intervals associated with variables and the primary endpoint (5 year recurrence); Circles represent the hazard ratio and the horizontal bars extend from; abbreviations- Grade-Int, Grade intermediate; HR-Hormone receptor status.

**Figure 3 cancers-14-03916-f003:**
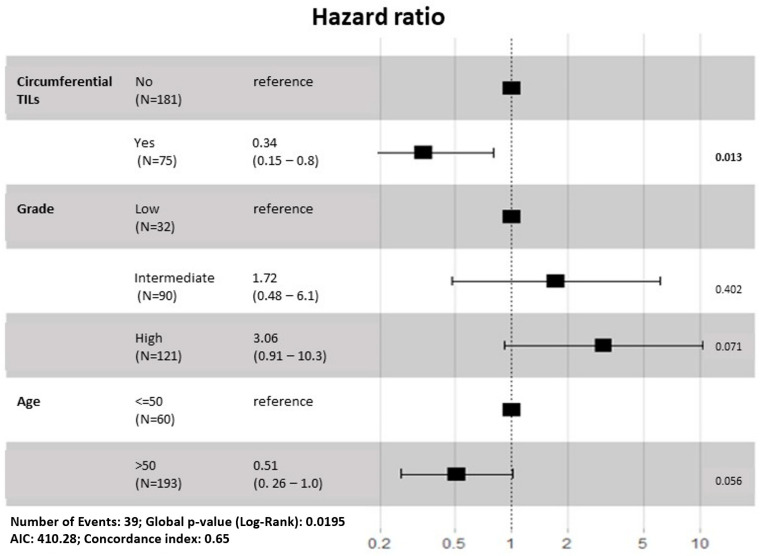
Forest plot of multivariable analysis of cohorts for tumor-infiltrating lymphocytes (TILs) assessments with clinical and histologic parameters. In total 3 variables were selected—age, grade, and circumferential TILs with significance (Wald test *p* = 0.01).

**Table 1 cancers-14-03916-t001:** Patient characteristics and association with TILs assessments for DCIS cohorts.

	Total (*n* = 256)
**Age**	
N-Miss	3
<=50	60 (23.7%)
>50	193 (76.3%)
**Size (mm)**	
N-Miss	50
<=20	131 (63.6%)
>20	75 (36.4%)
**Grade**	
N-Miss	13
Low	32 (13.2%)
Intermediate	90 (37.0%)
High	121 (49.8%)
**HR**	
N-Miss	86
Positive	137 (80.6%)
Negative	33 (19.4%)
**Recurrence**	
N	189 (73.8%)
Y	67 (26.2%)
**Lymphocyte**	
0–5%	101 (39.5%)
>5%	155 (60.5%)
**Touching TILs**	
0	214 (83.6%)
>0	42 (16.4%)
**Circumferential TILs**	
No	181 (70.7%)
Yes	75 (29.3%)
**Hotspot**	
No	160 (62.5%)
Dense	96 (37.5%)

## Data Availability

The data presented in this study are available upon request.

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
