# Peer review of "Tumor Infiltrating Lymphocytes in Multi-National Cohorts of Ductal Carcinoma In Situ (DCIS) of Breast"

_cancers, 2022, doi:10.3390/cancers14163916_

Round 1

Reviewer 1 Report

The authors conduct an interesting study focusing on tumor infiltrating lymphocytes and their association with recurrence among a multi-institutional/multi-ethnic cohort of women with DCIS. Prior studies identified that TIL touching the ducts circumferentially were an independent prognostic variable in DCIS. One rationale for the importance of TILs in tumor treatment is that given HLA genes dictate the adaptive immune response to neo-antigens, there may be racial/genetic differences that influence an individualʻs ability to mount an anti-tumor reponse. Given racial and ethnic disparities occur both in immune responses as well as DCIS, the authors conducted an analysis to evaluate the prognostication of TILs for ipsilateral breast cancer recurrence among a multi-institutional/multi-ethnic cohort.

The authors include four cohorts for assessment: Oxford, Singapore, Italy, and Mayo Clinic.

The authors assess TILs in the four cohorts and find the Singapore cohort had higher circumferential TILs and stromal hotspots. They then correlate TILs with second breast cancer outcomes and find circumferential TILs were associated with decreased risk of recurrence. The authors analyze second breast cancer events. They use multivariable CPH modeling to identify variables associated with second breast cancer events, yielding prognostic factors including: circumferential TILs, age, and hormone receptor positivity. The authors attempt to report the relationship between ethnicity and tumor infiltrating lymphocytes (circumferential or touching) and second breast cancer. They evaluate ethnicity for the Singapore, Mayo, and Italian cohorts, but not the Oxford cohort. They suggest in exploratory analyses that there were no significant differences in TILs or second breast cancer by Asian cohorts. They suggest that Asian ethnicity was associated with young diagnosis, larger size, and TILs.

In general, it is difficult to call this a multi-ethnic study given the lack of discrete self-reported racial/ethnic data reported. The only data on race is based on genetics, which may or may not correlate with reported race. Moreover, not all of the institutions were included in the race report. It appears the authors made assumptions on the patient’s race based on hospital, which would reflect more on geography (and therefore treatment practices) rather than ancestry. Moreover, because each institution has significant homogeneity by race, endpoints like survival are hard to interpret in the context of race because these treatment practices are not controlled for in this study. The studies do not appear to be controlled for treatment or location. This may significantly confound the analyses.

The paper does have multiple typos as detailed below. The figures need significant improvement for clarity. Many of the labels/legends are confusing/vague and should be rewritten for clarity. There are missing axis labels and there are inconsistencies between the legends and the actual figures. The figures should be cleaned up significantly for clarity. I do think the paper may be of utility and have provided some recommendations below:

MAJOR:

1.     Ln 146: If race was not self-identified and assumed, it would not be appropriate to call this “Ancestry”. For example, were all patients at Mayo Arizona self-reported White, this institution also treats patients who are Asian. It would be more appropriate to categorize this variable as Hospital/Institution. For example “Singapore Hospital” vs not “Asian ethnicity”. Further clarification is needed.

2.     Ln 163: Please describe in the methods whether proportional hazards assumptions were checked and if so how.

3.     Ln 168: Please either explain why stepwise variable selection was used over other methods or supply a citation.

4.     Ln 171: Please specify whether race data was explicitly collected and self reported. Please specify if race data was initially planned as a part of IRB? Was this self-reported or assumed race based on geographical location? It may be helpful to specify Asian or White race with terminology that most accurately reflects the cohort. If All Asian patients were ethnically Singaporean, then I would define the cohort in this way. Prior reports have disaggregated Asian by major ethnic populations and have shown differences in risk of developing ipsilateral/contralateral second breast cancer after DCIS among Asian and Pacific Islander subpopulations (PMID: 34668945)

5.     Ln 172: It is not clear if the “Asian” cohort is Singapore vs non-Singapore or if it all of the Asian from all 4 cohorts, or the genetic ancestry. Please be consistent with the definition throughout all analyses.

6.     Ln 177: Please report the median follow-up time for the whole cohort.

7.     Ln 194: What does this P value represent Singapore vs non-Singapore or the difference of all of the four cohorts?

8.     Ln 231: Please clarify if this endpoint is a binary yes or no for recurrence at 5 years?

9.     Ln 234: Please write in this first sentence for clarity which endpoint this is. CPH model for recurrence free survival or second breast cancer?

10.  Ln 239: Please be more clear in the legends what the endpoint is that is being assessed? Is this Recurrence free survival or second breast cancer?

11.  Ln 246: Please add a reference for the K13 model.

12.  Ln 311: Asians and Pacific Islanders (i.e. Native Hawaiians) do have different outcomes for developing second breast cancers after DCIS. Would recommend balancing any references to aggregated AA and PI data with more appropriate disaggregated AA and PI data showing the disparities pronounced in Pacific Islander populations (PMID: PMID: 34668945).

13.  Ln 323: Please address in the limitations 1) how these are not self-reported Racial groups and 2) why treatment location and treatment type were not used in the multivariable analyses, 3) discuss the potential bias for differences in pathology review methodology between Mayo/Italian, Singapore, and Oxford cohort. Why was there not a central review? What if anything ensured the mitigation of bias?

14.  Figure 1: I’m not sure what this correlation plot actually adds to the overall manuscript, possibly can add to supplements given its exploratory nature. Especially because everything seems to correlate. The data of the manuscript still stands on its own without Figure 1.

15.  Figure 2A: This figure is confusing. The endpoint is survival probability but the legend says 5 year recurrence. If this is survival probability, this is not a 5-year recurrence free survival endpoint (KM measures survival overtime not at one time point). Figure 2B. there are no axis labels.

16.  General: Throughout the paper the primary endpoint is referred to as “event”, “recurrence”, or “second breast cancer”. These terminologies are very confusing and should be streamlined throughout the paper so that the readers know which variables are being assessed.

MINOR:

1.     Ln 41: In the abstract, define what “good” and “bad” outcomes are. Is this survival? Recurrence?

2.     Ln 114-124: This text represents results not methods. Please rephrase this paragraph so that it reads as methods, not results. Please define more precisely the inclusion criteria for the study including the minimum follow-up as defined in the IRB protocol.

3.     Ln 118: Please describe in the results the breakdown of HR as ER and PR data. Add to the table; or if not in the table, include in the text.

4.     Ln 57. Define TNBC in first instance in text.

5.     Ln 15. “primarily treated with primarily by” typo

6.     Ln 67. Define DCIS only once in the first instance.

7.     Ln 82. Only need to define TILs once.

8.     Ln 95. Touching TILs were prognostic for which specific variable? PFS? OS? Recurrence?

9.     Ln 108: Please define definition of second breast cancer (i.e. timing from initial DCIS diagnosis).

10.  Ln 179: Please summarize the overall patient cohort in the first paragraph of the results including the number of patients, demographic info, and cancer characteristics. Usually in the first sentence.

11.  Table 1: Please add foot note to table defining abbreviations (HR, TILs).

12.  Figure 1: Please change labels in R throughout to reflect more meaningful and appropriate labels (e.g. “tilname.n”, “sig_level_f”, and “cor_dir”). Also the label looks like “Grade( Low)” if so should be fixed.

13.  Stats: Please be consistent throughout with number of decimal places. For example ln 206-209 please keep 3 decimals throughout.

14.  Ln 218: typo. CPH “models were”

15.  Ln 219: HR should be defined once

16.  Figure 2: Please clarify if this is univariable unadjusted

17.  Ln 235: Please consistent with use of “multivariable” not multivariate; Also consistent with “Cox” instead of cox.

18.  259: In general, the discussion would be improved by a one first paragraph summary of what the results show in the paper in a big picture sense.

19.  Ln 268: multi-centric or multiethnic?

Author Response

Comments and Suggestions for Authors

In general, it is difficult to call this a multi-ethnic study given the lack of discrete self-reported racial/ethnic data reported. The only data on race is based on genetics, which may or may not correlate with reported race. Moreover, not all of the institutions were included in the race report. It appears the authors made assumptions on the patient’s race based on hospital, which would reflect more on geography (and therefore treatment practices) rather than ancestry. Moreover, because each institution has significant homogeneity by race, endpoints like survival are hard to interpret in the context of race because these treatment practices are not controlled for in this study. The studies do not appear to be controlled for treatment or location. This may significantly confound the analyses.

We thank the reviewer for these comments. The primary goal of the study to analyze the association of TILs with adverse outcomes in DCIS. We used cohorts from 3 different continents that includes differences in race, ethnicity and treatments. To clarify the issue of lack of self-reported ethnic data, we have added information regarding both self-reported racial/ethnic data and genetic analyses. It is undoubtedly true that differences in patient cohorts including treatment given can confound outcomes. This is to be expected in retrospective cohorts and is in congruence with other published studies on this topic.

The paper does have multiple typos as detailed below. The figures need significant improvement for clarity. Many of the labels/legends are confusing/vague and should be rewritten for clarity. There are missing axis labels and there are inconsistencies between the legends and the actual figures. The figures should be cleaned up significantly for clarity. I do think the paper may be of utility and have provided some recommendations below:

We corrected the typos and clarified the figures and legends.

MAJOR:

  1. Ln 146: If race was not self-identified and assumed, it would not be appropriate to call this “Ancestry”. For example, were all patients at Mayo Arizona self-reported White, this institution also treats patients who are Asian. It would be more appropriate to categorize this variable as Hospital/Institution. For example “Singapore Hospital” vs not “Asian ethnicity”. Further clarification is needed.

The Race was self-identified by the patient at the time of admission to the respective institutions (except for Oxford).  We further analyzed Ancestry using a panel of genes (need to provide the details for the Illumina assay) and described the criteria used to classify the subjects in to distinct groups. Additional information regarding self-reported ethnicity is added to the results section, lines 174-176.

  1. Ln 163: Please describe in the methods whether proportional hazards assumptions were checked and if so how.

Proportional hazards assumptions were checked based on the scaled Schoenfeld residuals [P. Grambsch and T. Therneau (1994), Proportional hazards tests and diagnostics based on weighted residuals. Biometrika, 81, 515-26.]

  1. Ln 168: Please either explain why stepwise variable selection was used over other methods or supply a citation.

The stepwise variable selection procedure in which the choice of predictive variables is carried out by an automatic procedure was used. (Miller 2002; Hicks, et al. 2020; Chen, et al. 2019; Roussel, et al. 2021; Doshi, et al. 2020; Zhang, et al. 2020; Armstrong, et al. 2018). This heuristic procedure consists of a series of alternating forward selection and backward elimination steps, and ultimately selects a subset of variables to be included in the Cox-PH model.

  • Miller, A. (2002). Subset Selection in Regression (2nd ed.). Chapman and Hall/CRC. https://doi.org/10.1201/9781420035933
  • Hicks, C.W., Canner, J.K., Mathioudakis, N., Lippincott, C., Sherman, R.L. and Abularrage, C.J., 2020. Incidence and risk factors associated with ulcer recurrence among patients with diabetic foot ulcers treated in a multidisciplinary setting. Journal of Surgical Research246, pp.243-250.
  • Chen, T., Li, X., Li, Y., Xia, E., Qin, Y., Liang, S., Xu, F., Liang, D., Zeng, C. and Liu, Z., 2019. Prediction and risk stratification of kidney outcomes in IgA nephropathy. American journal of kidney diseases74(3), pp.300-309.
  • Roussel, E., Peeters, E., Vanthoor, J., Bozzini, G., Muneer, A., Ayres, B., Sri, D., Watkin, N., Bhattar, R., Parnham, A. and Sangar, V., 2021. Predictors of local recurrence and its impact on survival after glansectomy for penile cancer: time to challenge the dogma?. BJU international127(5), pp.606-613.
  • Doshi, R.P., Chen, K., Wang, F., Schwartz, H., Herzog, A. and Aseltine, R.H., 2020. Identifying risk factors for mortality among patients previously hospitalized for a suicide attempt. Scientific reports10(1), pp.1-9.
  • Zhang, C., Zhang, Z., Zhang, G., Zhang, Z., Luo, Y., Wang, F., Wang, S., Che, Y., Zeng, Q., Sun, N. and He, J., 2020. Clinical significance and inflammatory landscapes of a novel recurrence-associated immune signature in early-stage lung adenocarcinoma. Cancer letters479, pp.31-41.
  • Armstrong, A.J., Lin, P., Higano, C.S., Sternberg, C.N., Sonpavde, G., Tombal, B., Templeton, A.J., Fizazi, K., Phung, D., Wong, E.K. and Krivoshik, A., 2018. Development and validation of a prognostic model for overall survival in chemotherapy-naïve men with metastatic castration-resistant prostate cancer. Annals of Oncology29(11), pp.2200-2207.

  1. 4.    Ln 171: Please specify whether race data was explicitly collected and self-reported. Please specify if race data was initially planned as a part of IRB? Was this self-reported or assumed race based on geographical location? It may be helpful to specify Asian or White race with terminology that most accurately reflects the cohort. If All Asian patients were ethnically Singaporean, then I would define the cohort in this way. Prior reports have disaggregated Asian by major ethnic populations and have shown differences in risk of developing ipsilateral/contralateral second breast cancer after DCIS among Asian and Pacific Islander subpopulations (PMID: 34668945).

Ethnicity and Race were predefined and part of the IRB approval. Similarly, the DNA analysis for determination of ancestry was also approved by the IRB. We attempted to separate the populations from each cohort by genetic subsets, however the numbers are too small to draw any meaningful conclusions. See lines 333-336

  1. 5.   Ln 172: It is not clear if the “Asian” cohort is Singapore vs non-Singapore or if it all of the Asian from all 4 cohorts, or the genetic ancestry. Please be consistent with the definition throughout all analyses.

The Asian cohort consists of both Singaporean and non-Singaporean patients. We did not have DNA on the Oxford cohort but only the Mayo cohort had Asian patients of non-Singaporean origin, based on DNA analysis.

  1. Ln 177: Please report the median follow-up time for the whole cohort.

Mean, standard deviation, range, and number of missing values are added to Table 1. Follow up time for each recurrent/non-recurrent patient groups are included in Supplementary Table 2.

  1. 7.    Ln 194: What does this P value represent Singapore vs non-Singapore or the difference of all of the four cohorts?

We clarified that the p-value is evaluating if the distribution of TILs score is cohort dependent. We used Fisher’s exact test as we described in method section.

  1. Ln 231: Please clarify if this endpoint is a binary yes or no for recurrence at 5 years?

A binary endpoint (yes or no) endpoint was used for the analysis.

  1. 9.     Ln 234: Please write in this first sentence for clarity which endpoint this is. CPH model for recurrence free survival or second breast cancer?

We clarified that the endpoint of the multivariable analysis is the presence or absence of second breast cancer event within 5 years.

  1. Ln 239: Please be more clear in the legends what the endpoint is that is being assessed? Is this Recurrence free survival or second breast cancer?

We clarified that the endpoint of the multivariate analysis is recurrence within 5 years.  

  1. Ln 246: Please add a reference for the K13 model.

Added as suggested.

  1. Ln 311: Asians and Pacific Islanders (i.e. Native Hawaiians) do have different outcomes for developing second breast cancers after DCIS. Would recommend balancing any references to aggregated AA and PI data with more appropriate disaggregated AA and PI data showing the disparities pronounced in Pacific Islander populations (PMID: PMID: 34668945).

We neither have any patients nor expertise regards disease in Pacific Islanders and therefore would defer the comments to a more learned group. However, based on the reviewer’s comment, we have added a this to the discussion lines 333-336.

  1. Ln 323: Please address in the limitations
  2. a) how these are not self-reported Racial groups.

We report self-reported as well as genetic groups in this work.

  1. b) why treatment location and treatment type were not used in the multivariable analyses.

There are differences in the treatment based on location and even within each location. In addition, as the study was retrospective in nature, the treatment type was not available on all patients, particularly from the Oxford cohort. Lastly, the sample size is too small to incorporate treatment type into analysis.

  1. c) discuss the potential bias for differences in pathology review methodology between Mayo/Italian, Singapore, and Oxford cohort. Why was there not a central review? What if anything ensured the mitigation of bias?

We conducted a central review of the pathology material. The only differences in the review process consists of TMA (Oxford) versus whole slides (Mayo and Singapore).

  1. Figure 1: I’m not sure what this correlation plot actually adds to the overall manuscript, possibly can add to supplements given its exploratory nature. Especially because everything seems to correlate. The data of the manuscript still stands on its own without Figure 1.

We have believe this image add to the manuscript, however, if the reviewer insists we can remove it.

  1. Figure 2A: This figure is confusing. The endpoint is survival probability but the legend says 5 year recurrence. If this is survival probability, this is not a 5-year recurrence free survival endpoint (KM measures survival overtime not at one time point). Figure 2B. there are no axis labels.

The end point of the KM plots was 5year recurrence. We changed the y-axis label to recurrence free probability to be accurate. Also, add ‘Hazard ratio on 5year recurrence’ is added to the figure 2B.

  1. General: Throughout the paper the primary endpoint is referred to as “event”, “recurrence”, or “second breast cancer”. These terminologies are very confusing and should be streamlined throughout the paper so that the readers know which variables are being assessed.

            We thank the reviewer for pointing this out. We have modified this in the new version.

MINOR:

  1. Ln 41: In the abstract, define what “good” and “bad” outcomes are. Is this survival? Recurrence?

The terms are defined in reference to Second breast cancer event at 5 years. The language is modified to reflect this more accurately. 

  1. Ln 114-124: This text represents results not methods. Please rephrase this paragraph so that it reads as methods, not results.

These are the descriptions of the cohort and were available at the beginning of the study so are technically not results of the study. However, in view of the reviewers comment, this section is relocated to the results section.

  1. Please define more precisely the inclusion criteria for the study including the minimum follow-up as defined in the IRB protocol.

The basic criteria for inclusion was a confirmed diagnosis of DCIS and a followup data that included a) an in breast second event or b) follow-up of 5 years without any events.

  1. Ln 118: Please describe in the results the breakdown of HR as ER and PR data. Add to the table; or if not in the table, include in the text.

Modified as suggested.

  1. Ln 57. Define TNBC in first instance in text.

Modified as suggested.

  1. Ln 15. “primarily treated with primarily by” typo

Modified as suggested.

  1. Ln 67. Define DCIS only once in the first instance.

Modified as suggested.

  1. Ln 82. Only need to define TILs once.

Modified as suggested.

  1. Ln 95. Touching TILs were prognostic for which specific variable? PFS? OS? Recurrence?

Modified as suggested.

  1. Ln 108: Please define definition of second breast cancer (i.e. timing from initial DCIS diagnosis).

Modified as suggested. Second event was defined as ipsilateral event occurring more than 6 months after the initial diagnosis.

  1. Ln 179: Please summarize the overall patient cohort in the first paragraph of the results including the number of patients, demographic info, and cancer characteristics. Usually in the first sentence.

We re-located the paragraph summarizing the overall patient cohort, from the method section to result section

  1. Table 1: Please add foot note to table defining abbreviations (HR, TILs).

Instead of foot notes, we have added directly into the tables.

  1. Figure 1: Please change labels in R throughout to reflect more meaningful and appropriate labels (e.g. “tilname.n”, “sig_level_f”, and “cor_dir”). Also the label looks like “Grade( Low)” if so should be fixed.

Modified as suggested.

  1. Stats: Please be consistent throughout with number of decimal places. For example ln 206-209 please keep 3 decimals throughout.

Modified as suggested.

  1. Ln 218: typo. CPH “models were”

Modified as suggested.

  1. Ln 219: HR should be defined once

Modified as suggested.

  1. Figure 2: Please clarify if this is univariable unadjusted

Modified as suggested.

  1. Ln 235: Please consistent with use of “multivariable” not multivariate; Also consistent with “Cox” instead of cox.

Modified as suggested.

  1. 259: In general, the discussion would be improved by a one first paragraph summary of what the results show in the paper in a big picture sense.

Modified as suggested.

  1. Ln 268: multi-centric or multiethnic?

The study is both multicentric as well as multiethnic as clarified in the modified text.

Reviewer 2 Report

Comment 1.

The authors should investigate the relationship between the tumor mutational burden degree in DCIS and the TILs (the stromal, touching, circumferential and hotspots). However, the authors did not perform the mutation analysis by sequencing the DCIS samples. That is, which kind of tumor-infiltrating lymphocytes did respond to the increment of tumor mutational burden in the DCIS samples, this is key point underlying the main research objective of the authors. The authors should describe in the revised discussion section that such key point will be resolved in their future research. 

Author Response

Review2

The authors should investigate the relationship between the tumor mutational burden degree in DCIS and the TILs (the stromal, touching, circumferential and hotspots). However, the authors did not perform the mutation analysis by sequencing the DCIS samples. That is, which kind of tumor-infiltrating lymphocytes did respond to the increment of tumor mutational burden in the DCIS samples, this is key point underlying the main research objective of the authors. The authors should describe in the revised discussion section that such key point will be resolved in their future research. 

We thank the reviewer for his/her comment. We agree that having mutational burden would be valuable in any series studying immune infiltrations. However, there are 2 factors that prevented us from performing this analysis. The incidence of Tumor Mutational burden (TMB) in breast cancer, even invasive cancer, is very low; less than 5% of cases have a TMB > 10. We presume that TMB in DCIS will be similar if not lower. The second reason is the high cost of analysis. We attempted to get funding for the analysis, but this was denied by granting agencies on the grounds of low incidence of TMB in breast cancer.

Round 2

Reviewer 1 Report

The authors have addressed my many points mostly adequately. These corrections have improved the overall quality of the paper and have provided needed clarity.

I would recommend a thorough read through for formatting and spelling. For example. Results are labeled as 3.1, 3.1.1, 3.1.3, 3.14.

Figures still need to be improved before publication: Any abbreviation in a figure or table should be defined or removed if unnecessary. Figure 1 for example, "sig_level_f" should be changed to what it actually represents F or P statistic. Likewise for "cor_dir". These are defaults from R and should be adjusted to be meaningful to the authors.

Author Response

We have modified the figures as suggested and done a thorough read of the manuscript to eliminate any typographical errors.